# Describing and Characterizing the Literature Regarding Umbilical Health in Intensively Raised Cattle: A Scoping Review

**DOI:** 10.3390/vetsci9060288

**Published:** 2022-06-11

**Authors:** Matthew B. Van Camp, David L. Renaud, Todd F. Duffield, Diego E. Gomez, William J. McFarlane, Joanne Marshall, Charlotte B. Winder

**Affiliations:** 1Department of Population Medicine, University of Guelph, Guelph, ON N1G 2W1, Canada; vancampm@uoguelph.ca (M.B.V.C.); renaudd@uoguelph.ca (D.L.R.); tduffiel@uoguelph.ca (T.F.D.); wmcfarla@uoguelph.ca (W.J.M.); jmarsh12@uoguelph.ca (J.M.); 2Department of Clinical Studies, University of Guelph, Guelph, ON N1G 2W1, Canada; dgomezni@uoguelph.ca

**Keywords:** cattle, navel, morbidity, omphalitis, umbilicus

## Abstract

The objective of this scoping review was to describe and characterize the existing literature regarding umbilical health and identify gaps in knowledge. Six databases were searched for studies examining umbilical health in an intensively raised cattle population. There were 4249 articles initially identified; from these, 723 full text articles were then screened, with 150 articles included in the review. Studies were conducted in the USA (*n* = 41), Brazil (*n* = 24), Canada (*n* = 13), UK (*n* = 10), and 37 additional countries. Seventeen were classified as descriptive, 24 were clinical trials, and 109 were analytical observational studies. Umbilical outcomes evaluated in descriptive studies were infection (*n* = 11), parasitic infection (*n* = 5), and hernias (*n* = 2). Of the clinical trials, only one examined treatment of navel infections; the remainder evaluated preventative management factors for navel health outcomes (including infections (*n* = 17), myiasis (*n* = 3), measurements (*n* = 5), hernias (*n* = 1), and edema (*n* = 1)). Analytical observational studies examined risk factors for umbilical health (*n* = 60) and umbilical health as a risk factor (*n* = 60). Studies examining risk factors for umbilical health included navel health outcomes of infections (*n* = 28; 11 of which were not further defined), hernias (*n* = 8), scoring the navel sheath/flap size (*n* = 16), myiasis (*n* = 2), and measurements (*n* = 6). Studies examining umbilical health as a risk factor defined these risk factors as infection (*n* = 39; of which 13 were not further defined), hernias (*n* = 8; of which 4 were not further defined), navel dipping (*n* = 12), navel/sheath scores as part of conformation classification for breeding (*n* = 2), measurements (*n* = 3), and umbilical cord drying times (*n* = 2). This review highlights the areas in need of future umbilical health research such as clinical trials evaluating the efficacy of different treatments for umbilical infection. It also emphasizes the importance for future studies to clearly define umbilical health outcomes of interest, and consider standardization of these measures, including time at risk.

## 1. Introduction

Neonatal calf management practices are of vital importance to producers due to the susceptibility of newborn calves to developing disease. It was reported that 34% to 39% of dairy calves had at least one case of disease during the pre-weaned period, of which 90% received antimicrobial therapy [1,2]. Furthermore, a study examining morbidity in the veal industry reported that 25% of calves received at least one disease treatment from arrival to slaughter [3]. Additionally, 5% to 11% of calves died during the pre-weaning period [2,4]. Therefore, producers and researchers need to continue working together to improve animal welfare by reducing disease and mortality in calf populations.

A large body of literature has examined the benefits of providing high-quality calf care in early life. Specifically, research has highlighted the importance of housing and ventilation [5,6,7], nutrition [8], and colostrum management [9] to prevent neonatal diseases. Likewise, research has documented the major risk factors associated with the development of respiratory disease and calf diarrhea [2,10] as well as the consequences of developing these diseases [11,12,13]. However, knowledge gaps remain regarding the risk factors associated with umbilical health and understanding the impact that umbilical health has on calf productivity, health, and welfare.

The umbilical cord is the connection between the cow and her fetus that allows the passing of nutrients and oxygen, and the elimination of fetal waste products [14]. The cord comprises two arteries, a vein, and the urachus [14,15]. Following parturition, the umbilical cord dries and detaches leaving a structure which is known as the navel [14,15]. During this drying period, the umbilical cord is open to the surrounding environment, allowing pathogens to enter the umbilicus, colonize the tissues, and cause disease [14]. Once a pathogen colonizes the umbilical cord, it can cause an infection triggering an immune response [14,15]. Infection of any or all the umbilical structures is known as omphalitis [16]. There is a wide discrepancy in the reported incidence of omphalitis in calves.

Older literature has indicated that 1.3% [17] to 14% [18] of dairy calves develop an umbilical infection; however, newer research suggests that omphalitis might occur at a much higher incidence. Specifically, Johnson et al. [19] reported a prevalence of 28.7%, whereas Renaud et al. [20] found a 26% prevalence in dairy calves. One possible reason for the discrepancy in frequencies are the differences in disease definitions used in earlier and recent studies. Svensson et al. [17] and Virtala et al. [18] both used binary disease definitions; however, Johnson et al. [19] and Renaud et al. [20] each used the same categorical scoring system from 0 to 3 to diagnose infection that was adapted from Fecteau et al. [21].

There are also discrepancies within the current research regarding measures to improve umbilical health. Umbilical dip is thought to be an important preventative measure, with 40% of Canadian dairy farmers and 61% of French dairy farms using some form of navel dip [22,23]. However, despite the frequent use of umbilical dips, there remain conflicting results as to their effectiveness [24,25,26,27]. Other risk factors commonly linked to umbilical infections include the type of bedding used [28] and failed transfer of passive immunity (FTPI). A link between FTPI and increased calf morbidity and mortality is well established [29]; however, no association between FTPI with risk of umbilical infection was documented [19,30,31].

A scoping review seeks to collect and disseminate a wide scope of evidence on a topic [32]. Scoping reviews are generally performed to achieve one or more of the following: (1) to examine the current extent of the research; (2) to determine the value of performing a systematic review; (3) to summarize the current research findings and definitions; and (4) to identify current knowledge gaps [32]. This approach differs from a systematic review in that it does not summarize the literature to provide an answer for a specific question, nor does it evaluate the quality of evidence [33].

The objective of this scoping review was to describe and characterize the existing literature regarding umbilical health and to highlight current gaps in knowledge. This includes providing an overview of descriptive studies, clinical trials, and observational studies examining risk factors associated with umbilical health and the consequences of poor umbilical health.

## 2. Materials and Methods

### 2.1. Protocol and Registration

This review is reported using PRIMSA-Scr Reporting Guidelines [34]. The protocol for this scoping review can be found at: http://hdl.handle.net.subzero.lib.uoguelph.ca/10214/17821 (accessed on 3 March 2020).

### 2.2. Eligibility Criteria

Sources for this scoping review were drawn from primary research studies including observational and experimental studies that were published in English; case reports and case series studies were excluded. The target population was limited to cattle on dairy, veal, feedlot, cow–calf, or dairy beef operations (small holder or dual purpose animal operations were excluded). This included articles on cattle of any sex or age, with no geographic or date of publication limitations. Studies examining umbilical surgeries that did not describe a measure of umbilical health as an outcome were excluded. In addition, any of the studies performed pre-partum (on the fetus, e.g., examining umbilical blood flow), as well as any articles describing rare birth abnormalities (e.g., omphalocele) were excluded. Thus, the scoping review included publications examining an aspect of umbilical health including the prevalence or incidence of umbilical infections or abnormalities, upstream risk factors for umbilical health, or umbilical health as a risk factor itself.

### 2.3. Information Sources

The following databases were accessed online through the University of Guelph McLaughlin library: CAB Direct (via CAB Interface), SCOPUS, ProQuest dissertation and thesis (via ProQuest), Agricola (via ProQuest), Medline (via Ovid), and Science Citation Index Expanded (SCI-EXPANDED), Conference Proceedings Citation Index-Science (CPCI-S), and Emerging Sources Citation Index (ESCI) (via Web of Science).

### 2.4. Literature Search

Search terms were designed prior to literature screening and aimed to maximize sensitivity. Full terms and results from the test search are shown in Table 1. Search results from all databases were conducted on the 8th April 2020 and yielded 6815 unique records. Searches were then transferred to the reference data management software Endnote (Clarivate, Philadelphia, PA, USA). Following reference uploading, duplicate studies from all of the databases were documented and removed before screening via DistillerSR (Evidence Partners, Ottawa, ON, Canada).

### 2.5. Title and Abstract Screening

Two reviewers independently read the title and abstracts of the literature identified using the search terms and determined if they were appropriate to include in the scoping review. Each reviewer examined each article independently using the listed questions below:Is the title/abstract in English;Does the title/abstract describe a primary research article;Does the title/abstract include any aspects of umbilical and/or navel health in cattle?

Each reviewer (MBVC and WJM) then answered “yes”, “no”, or “unclear” to each question. Articles that received a “yes” or “unclear” to all three of the questions by both reviewers were included in the next phase of full text screening. Articles that had an agreed upon “no” to any question were excluded. When reviewers disagreed on inclusion or exclusion of an article, they resolved the conflict via consensus; all of the unresolved disagreements were resolved by a third party (DLR or CBW). A pre-test was completed on the first 100 articles to ensure they were accurately and consistently applying the study selection criteria.

### 2.6. Full Text Screening

After the title and abstract screening was completed, the included studies underwent full text screening. Each reviewer (MBVC and JM) independently read the full text and answered the questions below with either a “yes” or “no”:Is the full text available;Is the full text in English;Is the full text > 500 words;Is the full text a primary research article describing an observational or experimental study (including descriptive studies, controlled trials and observational studies–cohort/case control/cross sectional);Does the full text describe an examination of the prevalence or incidence of umbilical health or abnormalities, examination of risk factors of umbilical health (including intervention studies), or examination of umbilical health as a risk factor?

Articles that were given a “yes” to all of the questions by both reviewers were included in the scoping review. Articles that received an agreed upon “no” to any or all of the questions were excluded from the scoping review. All of the full text exclusions and reasons for exclusion were reported. Disagreement by the two reviewers for any of the question(s) were resolved via consensus and any unresolved conflicts were determined by a third party (DLR or CBW). A pre-test of the first 10 full text articles was completed to ensure that they were accurately and consistently applying the study selection criteria.

### 2.7. Data Extraction

Following full text screening, both reviewers (MBVC and JM) extracted data from all of the articles independently in duplicate. A pre-test was completed on the first five articles to ensure clarity of consensus and any unresolved conflicts were resolved by a third party (DLR or CBW) similar to the abstract and full text screening process.

The following information was extracted at the study level: authors; year of publication; year of conduct; country of study conduct; sample size; age, breed, and sex of study population; type of farm (research or commercial); and type of study (descriptive, clinical trial, or analytical observational study). For the descriptive studies, the prevalence or incidence of umbilical infections was recorded, as well as the case definition including time at risk. Analytical studies not meeting the eligibility requirements but providing descriptive umbilical health information were included as descriptive studies. Analytical studies that contained descriptive information that fit within the eligibility criteria were counted solely as observational studies and they were not recorded as descriptive studies. For clinical trials, a full description of all of the intervention groups was recorded, as well as a full description of all of the outcomes pertaining to umbilical health, including the case definition and time at risk. For the analytical observational studies examining risk factors for umbilical health, all of the exposures measured were listed, and a full description of all of the outcomes pertaining to umbilical health were recorded (case definition and time at risk). For the analytical observational studies where umbilical health was assessed as a risk factor, this was fully described (case definition and age at assessment), and all of the outcomes measured were listed. In cases where an observational analytical study examined both risk factors of umbilical health and umbilical health as a risk factor, they were recorded as one article but they had data extracted for both categories.

## 3. Results

### 3.1. Synthesis of Results

Of the initial 4249 articles identified for title and abstract screening, 723 articles were included in full text screening. Following full text screening, 150 articles were included in the data extraction phase. Extracted data were then moved from DistillerSR into a Microsoft Excel spreadsheet for the descriptive results to be characterized within this scoping review. The flow of studies through the screening process, including reasons for the full text exclusions, are shown in Figure 1.

Table 2 highlights the study types and ranges in dates of publication (1931 to 2020). Studies were conducted in 41 countries, including the USA (*n* = 41), Brazil (*n* = 24), Canada (*n* = 13), UK (*n* = 10), and Australia (*n* = 9) (Figure 2). Most of the studies were observational (109/150; 73%), followed by clinical trials (24/150; 16%) and descriptive studies (17/150; 11%). For additional information on all of the included studies, see the Supplemental Data Tables’ document available at https://doi.org/10.5683/SP2/M4J00A (accessed on 3 March 2020).

### 3.2. Descriptive Studies

The descriptive studies were conducted in the USA (4/17; 24%), Canada (2/17; 12%), Ethiopia (2/17; 12%) and one each in Australia, Bangladesh, Brazil, Germany, Guatemala, Indonesia, Iraq, Jamaica, and Sweden. The date of these publications ranged from 1946 to 2018, with most published after 2000 (10/17; 59%). Commercial herds were used in the majority of the studies (14/17; 82%), of which dairy (5/14; 36%) was the most common herd type, followed by cattle at an abattoir/slaughter (4/14; 26%), auction (1/14; 7%), dairy calf raiser (1/14; 7%), veal farms (1/14; 7%), a combination of dairy and beef cattle (1/14; 7%), and one study did not report the type of commercial herd used (1/14; 7%). In addition, one study was conducted in a research facility (1/17; 6%), and two did not report the study location (2/17; 12%). Within these descriptive studies, 29% (5/17) evaluated Holstein or Holstein Friesian, 12% (2/17) used mixed breeds, and 59% (10/17) did not report the breed of cattle used in the study. Many of the articles did not report the sex of the calves evaluated (5/22; 23%), however, of those that reported sex, 42% (7/17) used both sexes, 18% (3/17) used exclusively female, and 12% (2/17) used exclusively male. The age range of cattle included in the studies was from birth to 2 years of age, with 41% (7/17) of studies using cattle under one year of age, 18% (3/7) including ages ranging from birth to over one year of age, and the remainder of the studies (41% (7/17)) not reporting the age of the cattle.

Umbilical infections were the most common umbilical health outcome evaluated (11 of the 17 descriptive studies). Steerforth and Van Winden [36] examined post-mortem prevalence of omphalitis in calves 7 and 15 days of age, which they defined as “inflammation of any of the umbilical structures—including the umbilical arteries, umbilical vein, urachus or tissues”. Overall prevalence of omphalitis at post-mortem was 34% (64/187). The remaining 10 studies evaluating umbilical infections described the incidence (*n* = 5) or prevalence (*n* = 5) in live cattle. Of these, six did not include a definition of disease, two of the studies defined umbilical infection as an abnormal umbilicus requiring treatment with antimicrobials, Brisville et al. [37] used a combination of bacterial culture testing and attending clinician diagnosis, and Pempek et al. [38] used a 4-point scoring system, based on the size of the umbilicus. The prevalence of umbilical infections ranged from 0.3% [39] to 27% [38], whereas the incidence ranged from 0.003% [40] to 7.22% [41]. A unique study by Brisville et al. [36] examined umbilical health in cloned calves and found that 74.2% (23/31) had an enlarged navel at birth and 38.7% (12/31) of those calves developed an umbilical infection over the next two years.

Umbilical hernias were described in two of the included descriptive articles (2/17; 12%). Blakley [40] (who also described the incidence of omphalitis) included umbilical and scrotal hernias together and identified the incidence of hernias to be 11.7 per 1000 calves under 4 months of age at an auction barn over a 4-month period; meanwhile, Singh et al. [42] did not include a definition or time at risk for umbilical hernia, however, identified a prevalence of 2.2% (7/317).

Parasitic infections of the navel entailed the remaining descriptive articles that were included (5/17; 29%). Navel myiasis included the infestation of the navel by screwworm flies (2/5), roundworms of the genus *Onchocerca* (2/5), and other parasites (1/5). Each of these five studies determined the prevalence of navel myiasis, ranging from 5% [43] to 40% [44].

### 3.3. Clinical Trials

Clinical trials were conducted in the USA (9/24; 38%), Brazil (6/24; 25%), United Kingdom (3/24; 13%), Pakistan (2/24; 8%), Argentina (2/24; 8%), and one from each of Australia, Serbia, and Venezuela. The date of publication ranged from 1949 to 2018 with the majority being published after the year 2000 (17/24; 71%). Commercial herds were the most common study location used in clinical trials (16/24; 67%), followed by research facilities (4/24; 17%), and the remaining studies (4/24; 17%) did not report the location type used. Of the included studies, 50% (12/24) composed their entire study population from a single location.

The sample size of the included studies was broken down as 17% (4/24) enrolling 1 to 20 animals, 17% (4/20) enrolling 21 to 50 animals, 29% (7/24) enrolling 51 to 80 animals, 8% (2/24) enrolling 100 to 200 animals, 8% (2/24) enrolling 200 to 300 animals, and 17% (4/24; 17%) had more than 300 animals enrolled. The most common breed used was Holstein or Holstein Friesian (11/24; 46%), followed by a combination of breeds (8/24; 33%). Of the remaining five studies, four did not report the breed used and one used exclusively Bos Indicus cattle. Both sexes were used in 38% of the studies (9/24), 29% (7/24) used exclusively female cattle, 13% (3/24) used exclusively male cattle, and the remainder did not report the sex of the cattle (5/24; 21%). The time at risk of the cattle included in the trials varied from birth to lactating cattle; however, the majority (16/24; 67%) examined the calves under 3 months of age. The range of time at risk among the clinical trials is further described in Figure 3.

#### 3.3.1. Outcomes

The most common interventions applied to calves were aimed at preventing a negative navel health outcome (23/24; 96%), whereas only one study (1/24; 4%) examined treatment of navel infections. Specifically, Abbas et al. [45] analyzed the time to recover from navel ill where five different parental and topical treatments, as well as a positive and negative control, were evaluated, and outcomes were measured up to 5 days post treatment. Of the studies looking at interventions to prevent negative navel health outcomes, 74% (17/23) centered around umbilical infections. This included describing one of their umbilical health outcomes as navel ill (2/17; 12%), umbilical infection/omphalitis (11/17; 65%), or navel inflammation (4/17; 24%). In these studies, health outcomes were evaluated using different methods including using a clinical scoring system (4/17; 24%), determining health outcomes by a veterinarian (3/17; 18%), using a variety of clinical indicators such as pain, size, and temperature (4/17; 24%), and determining an umbilical infection solely by palpating the umbilical stump (1/17; 6%). The remaining studies evaluating umbilical infections as an outcome did not define how they tested for umbilical infections (5/17; 29%). Three of the studies examined navel myiasis as a measure of navel health (3/24; 13%) including the infestation of the navel with *Cochliomyia hominivorax* (1/3; 33%), screw worms (1/3; 33%), and unspecified navel myiasis (1/3; 33%). Another common umbilical health measurement taken was a quantitative measurement of the umbilicus itself (5/24; 21%) including the wound area and healing rate (3/5; 60%), days until the umbilical cord fell off (3/5; 60%), umbilical diameter and length (3/5; 60%), drying time of the umbilicus, and umbilical temperature (2/5; 40%). The remaining outcomes examined in the clinical trials were umbilical hernias (1/24; 4%) and umbilical edema (1/24; 4%).

#### 3.3.2. Interventions

The most common preventative interventions evaluated included umbilical dips or sprays (9/24; 38%). These included iodine-based dips (8/9; 98%), chlorhexidine (3/9; 33%), Navel Guard (Purified Water, Acidified Water, Isopropyl Alcohol, Surfactant, Citric Acid, FD&C Red #40 and FD&C Yellow #5; Vet One, Boise, ID (3/9; 33%)), nisin (1/9; 11%), trisodium citrate (1/9; 11%), chlorine (1/9; 11%), and antibacterial peptides (1/9; 11%). Of the studies analyzing umbilical dips, 4/9 (44%) used a negative control. Additionally, two of the studies evaluating iodine-based umbilical dips also applied a form of light therapy as a method of speeding up umbilical drying times and reducing infections. Studies also evaluated the use of orally administered treatments (11/24; 42%) including supplements (*n* = 4) (one each of fatty acids, zeolite, vitamin A, and lactoferrin and cinnamaldehyde), changes to the feeding/rearing system (*n* = 3), changes in the colostrum source (*n* = 3), and dry cow nutrition (*n* = 1). Of the remaining three trials, two used subcutaneous injection of ivermectin and one study by de Melo Barbieri et al. [46] compared a positive control group of a standard anti-parasite drug regimen to a new anti-parasite administration regimen designed by the authors.

### 3.4. Analytical Observational Studies (Risk Factors for Umbilical Health)

The observational studies examining risk factors for umbilical health were conducted in Brazil (14/60; 23%), USA (13/60; 22%), Canada (4/60; 7%), Germany (3/60; 5%), Australia (3/60; 5%), Russia (2/60; 3%), the Czech Republic (2/60; 3%), Argentina (2/60; 3%), and several other countries (15/60; 25%). The date of publication ranged from 1931 to 2020, although the majority (49/60; 82%) were published after 2000. Commercial farms were used in most of the studies (37/60; 62%), followed by studies that did not report the location (14/60; 23%), and those that used a research farm or hospital (9/60; 15%). Of these study locations, 18% (11/60) only used one location to gather information for their entire study.

Eight percent (5/60) of the studies enrolled 1 to 100 animals, 15% (9/60) enrolled 101 to 300 animals, 17% (10/60) enrolled 301 to 600 animals, 7% (4/60) enrolled 601 to 1000 animals, 28% (17/60) enrolled 1001 to 10,000 animals, and 17% (10/60) enrolled more than 10,000 animals. Five percent of the studies (3/60) did not report the sample size used. The breeds of cattle used included mixed breed (23/60; 38%), Holsteins or Holstein Frisian (11/60; 18%), Nelore (8/60; 13%), and ten other breeds each of which was used exclusively in a single study. Seven of the studies included did not describe the breed of cattle.

Regarding the sex of the cattle, 53% (32/60) of the studies used both sexes of cattle, whereas 18% (11/60) used exclusively female, 7% (4/60) used only male, and the remainder did not report the sex of cattle used in their study. The age range of the cattle used varied from birth to adult cattle (the oldest animals reported were 17 years of age). However, 53% (32/60) of the studies reported using only cattle under 1 year of age. The range of time at risk between the observational studies examining risk factors for umbilical health is further described in Figure 4.

#### 3.4.1. Umbilical Health Outcomes

The most common umbilical health outcome examined was umbilical disease (36/60; 60%). This included umbilical infections (28/36; 78%) of which (3/28; 46%) used a binary scoring system (diseased or non-diseased) based on a list of clinical signs such as purulent discharge, pain, and temperature, 7% (2/28) defined infection as an enlarged umbilical region and/or persistent urachus, one used a veterinary diagnosis (not further defined), and one used treatment with antibiotics as a proxy for infection. A total of 39% (11/28) of the studies did not describe how they defined umbilical infections. Umbilical hernias were also evaluated in 22% (8/36) of the studies. Definitions for umbilical hernias varied, with one study describing them as congenital umbilical hernias with no further definition, two of the studies [47,48] used a definition of “palpable openings in the umbilical region >1.5 cm were defined as umbilical hernias, even if no hernial sac was developed. Inflammation, abscesses or fistulae were excluded from the diagnosis”, two of the studies defined them as palpation of any umbilical sacculation, and the remaining three of the studies diagnosed umbilical hernias through the evaluation by a veterinarian (2/3; 66%) or a herd manager (1/3; 33%). Two of the studies (3%) examined navel myiasis including infestation of the navel with *Onchocerciasis microfillae* (1/2; 50%), and *Onchocera gutturosa* (1/2; 50%). Other umbilical health outcomes included scoring the navel sheath/flap size as a confirmation score (16/60; 27%), using a quantitative measurement such as navel length/depth/diameter (4/6; 66%), sheath/flap area (1/6; 17%), umbilical scar tissue (1/6; 17%), and umbilical cord drying time (1/6; 17%).

#### 3.4.2. Risk Factors Examined

The categories of potential risk factors for umbilical health outcomes included cattle characteristics (33/60; 55%), genetic indicators (22/60; 37%), rearing practices (14/60; 23%), morbidity (4/60; 7%), and clinicopathological (13/60; 22%). The cattle characteristics examined included sex (13/33; 39%), breed (17/33; 52%), weight (7/33; 21%), height/growth (4/33; 12%), age (5/33; 15%), body condition score (1/33; 3%), cross suckling habits (2/33; 6%), and navel characteristics (2/33; 6%). Genetic risk factors examined included heritability estimates/traits/values (11/22; 50%), genotype (8/22; 36%), breeding values (4/22; 18%), sire information (9/22; 41%), and dam information (9/22; 41%). Cattle rearing practices examined included herds that the cattle originated from (3/14; 21%), cloned calves (3/14; 21%), season of birth (5/14; 36%), distance to water sources (1/14; 7%), colostrum management practices (2/14; 14%), housing practices including rearing systems such as group housing and individual pens (5/14; 36%), calving pen management (3/14; 21%), region of birth (1/14; 7%), diet (1/14; 7%), and navel dipping (1/14; 7%). Previous morbidity was examined as a potential risk factor in four of the studies including health score or neonatal vitality score (2/4; 50%), joint ill (1/4; 25%), diarrhea (1/4; 25%), eye/nasal discharge (1/4; 25%), navel myiasis (1/4; 25%), and pre-eclampsia (1/4; 25%). Finally, clinicopathological factors were assessed in 21% (13/60) of the studies and included concentrations of total serum protein (4/13; 31%), serum IgG (5/13; 38%), albumin (1/13; 8%), gamma globulin (1/13; 8%), vitamin A (2/13; 15%), vitamin C (1/13; 8%), carotene (1/13; 8%), and other metabolic markers such as amino acids, fatty acid profile, cholesterol, and additional vitamins (4/13; 31%).

### 3.5. Analytical Observational Studies (Umbilical Health as a Risk Factor)

The observational studies examining umbilical health as a risk factor were conducted in the USA (15/60; 25%), Canada (7/60; 12%), New Zealand (6/60; 10%), Turkey (5/60; 8%), United Kingdom (4/60; 7%), Finland (4/60; 7%), Brazil (2/60; 3%), Algeria (2/60; 3%), Ethiopia (2/60; 3%), Slovakia (2/60; 3%), and seven other countries. The date of publication ranged from 1931 to 2020, although the majority (42/60; 70%) were published post 2000. Commercial locations were used in 77% (46/60) of the studies, whereas research or hospitals were used in eight (13%) of the studies, and six (10%) of the studies did not describe the type of herd used. Of the included studies, 15% (9/60) only used one location for their entire study population.

With respect to sample size, 28% (17/60) enrolled 1 to 100 animals, 18% (11/60) enrolled 101 to 300 animals, 13% (8/60) enrolled 301 to 600 animals, 10% (6/10) enrolled 601 to 1000 animals, and 25% (15/60) enrolled 1001 to 10,000 animals. Five percent (3/60) did not report the sample size used. The breeds of cattle were primarily a mix of several breeds (17/60; 28%) followed by exclusively using Holstein or Holstein Friesian (13/60; 22%), and not reporting the specific breed(s) used in their study (26/60; 43%). Approximately half of the studies used both sexes of cattle (27/60; 45%), whereas only 7 (12%) used exclusively male, 12 (20%) used exclusively female, and the remainder did not report the sex of cattle observed in their study (15/60; 25%). The age range used varied from birth to adulthood (the oldest animals reported were 4 years of age); however, the majority of the studies included cattle under one year of age (43/60; 72%). The range of time at risk in the observational studies examining umbilical health as a risk is further described in Figure 5.

#### 3.5.1. Umbilical Health Risk Factors

The most common category of umbilical health risk factor examined in the observational studies was umbilical disease (44/60; 73%). The risk factors that fell under the general category of umbilical disease mostly included a yes/no definition of umbilical infections (40/44; 91%). This included umbilical disease definitions based on a list of clinical signs such as purulent discharge, pain, and temperature (17/40; 43%), and those that scored the navel on a scale based on the severity of the symptoms such as size, purulent discharge, pain, and temperature (7/40; 18%). Another way that umbilical disease was categorized was based on the type of navel infection, such as a localized or extended infection (2/40; 5%). However, a third of the studies (13/40; 33%) did not define umbilical infections. Other umbilical outcomes examined were umbilical hernias (9/44; 20%), of which one (11%) used an ultrasound to make the diagnosis, one (11%) determined umbilical hernia using previous veterinarian hospital records, three (33%) used binary criteria such as palpations >1 cm, and the remaining four (44%) of the studies did not define their method of diagnosis. A total of 12 (20%) of the studies analyzed the effects of navel dipping as a risk factor for umbilical infection/health, 2 (3%) of the studies used a navel/sheath score as a conformation classification to assess breeding characteristics, 1 (2%) study evaluated the umbilical cord thickness, 2 (3%) of the studies evaluated navel flap size, and the final 2 (3%) studies followed umbilical cord drying times.

#### 3.5.2. Outcomes

The general categories of outcomes from the observational studies examining umbilical health as a risk factor were mortality (22/60; 37%), morbidity (19/60; 32%), cattle characteristics (12/60; 20%), physiological measures (6/60; 10%), and blood serum markers (13/60; 22%). Additionally, four of the studies looked at carcass condemnation and wastage at slaughter. Of the 19 included articles analyzing morbidity, the primary diseases examined included diarrhea (6/19; 32%), bovine respiratory disease (5/19; 26%), bacteremia/sepsis (5/19; 26%), and a mixture of several physiological markers for morbidity which were often described as a “clinical outcome” or “clinical score” (4/19; 21%). Other morbidity outcomes examined in just a single article included cecal infarction, joint ill, fecal shedding of *Salmonella*, and antibiotics used to treat disease. Cattle characteristics described a variety of measurement including weight (4/15; 33%), body size (4/12; 33%), sheath depth (1/12; 8%), appetite (1/12; 8%), breeding characteristics (1/12; 8%) and age estimation (1/12; 8%). Behavioral characteristics such as the number of lying bouts and an approach test were analyzed in three of the studies and sexual behavior such as number of mounts and libido score were assessed in three additional studies. The physiologic measures examined (6/60; 10%) included heart rate (2/6; 33%), respiratory rate (3/6; 50%), artery diameter (1/6; 17%), and temperature (3/6; 50%); however, many of these articles used these calf characteristics as quantitative markers of morbidity, as described previously via “clinical scores”. The final general category of outcomes analyzed were blood serum markers which included articles analyzing serum total protein (2/13; 15%), acute phase proteins (4/13; 31%), and other metabolic markers such as amino acids, fatty acid profile, cholesterol, vitamins, and minerals (8/13; 62%). 

## 4. Discussion

This scoping review provides an overview of the literature surrounding umbilical health in intensively raised cattle. The included studies were categorized into three general categories, descriptive, clinical trials, and observational studies, the latter of which were further divided into observational studies examining risk factors of umbilical health and observational studies examining umbilical health as a risk factor. Despite the number of studies included, there remains a lack of knowledge regarding treatments for umbilical disease, which was only evaluated in a single clinical trial.

In addition, a notable theme that was identified throughout this review was the lack of consistent anatomical terms and definitions for umbilical health. In many cases, no clear definition or methodology was provided to classify disease. Within this review, a wide range of terms were used to describe umbilical infections. Navel ill, navel infection, omphalitis, omphalophlebitis, omphaloarteritis, patent urachus, and navel abscesses appeared to be used interchangeably for the same disease. This was very common, that the outcomes and risk factors described as such were classified under the general term “umbilical infection” in this review, because of the difficulties in distinguishing and separating them from each other. For example, omphalophlebitis is defined specifically as an infection of the umbilical vein [14] but many of the studies that used this term failed to distinguish specific infection of the umbilical vein as opposed to other umbilical structures. Therefore, it is important for researchers to use standardized terms to minimize the variability within the literature and prevent misinterpretation of results. This review also highlighted the lack of reported definitions used for umbilical health outcomes. A possible explanation for the lack of definitions is that umbilical health outcomes were not often considered a primary outcome in the reviewed studies, which may have led to poor reporting of umbilical health definitions. Anecdotally, it appeared that respiratory disease and diarrhea definitions were more thoroughly reported and were explored further compared to umbilical-related health outcomes. Nonetheless, providing adequate definitions is imperative for future researchers who wish to conduct a meta-analysis or systematic review on umbilical health-related questions, as accessing the quality of the literature and reproducibility of their research findings is a part of these reviews [33].

In future umbilical health research, greater use of consistent, repeatable outcomes should be employed. Researchers in this area could consider the creation of a standardized core outcome set for umbilical health in calves. This would improve comparability of the studies using these standardized outcome definitions, and allow reviewers and readers to better determine the overall treatment or intervention efficacy [49,50]. In turn, by creating and using standardized outcomes, there would be less potential for research waste from the studies that have outcomes that cannot be compared or contrasted to other results [51]. Currently, resources, such as the Core Outcome Measures in Effectiveness Trials Initiative [50], can be used to help in the creation of this standardized outcome reporting.

Reporting guidelines could also aid in strengthening future literature surrounding umbilical health. Specifically, the Reporting Guidelines for Randomized Controlled Trials for Livestock and Food Safety (REFLECT) [52] and the Strengthening the Reporting of Observational Studies in Epidemiology-Veterinary (STROBE-Vet) [53] statements specify what information is critical to report when conducting randomized controlled trials and observational studies, respectively. By using these guidelines, researchers increase the reproducibility of their work, as well as improve the ability to appraise the risk of bias by readers and reviewers [52,53]. This includes reporting all of the descriptive information regarding their outcomes and risk factors to allow other researchers to reproduce the exact frequency of testing, the number of observers, or training requirements that could increase the validity or precision of the outcomes [52,53]. Furthermore, intervention studies that fail to report key design features, such as randomization and blinding, are more likely to have a larger effect measure than those that do report such features [54,55,56]. Poor reporting of key study design features is not uncommon in animal health literature [57]. The creation of standardized outcomes for umbilical health in calves and comprehensive reporting in these studies can improve the utility of the data generated [33,53].

A limitation of the literature described In this review was the lack of intra- or inter-observer reliability testing completed. Many of the studies had multiple observers but very few of the studies evaluated the intra- or inter-observer reliability of the test and/or scoring system. This is important, especially for the studies using multiple observers and a scoring system because the reader is unable to determine how accurate the results are. A study conducted by Buczinski et al. [58] reported a moderate level of reliability between the observers evaluating a diagnostic test for bronchopneumonia despite using the same diagnostic test. Therefore, studies should report whether the diagnostic test used for detection of an outcome depends on the individual conducting it and how repeatable the results of this test are when performed by the same individual multiple times [59].

This review identified a wide range in the reported incidence and prevalence of umbilical health outcomes. In the descriptive studies evaluating umbilical infections, the prevalence ranged from 0.3% to 26.7% and the incidence ranged from 0.003% to 7.22%. The wide range of prevalence and incidence could be, at least in part, due to the large variation in disease definitions and because the variation in populations of interest with many different breeds, age ranges, times at risk, geographic locations, and sexes of cattle used in the studies. The length of time that the subjects were followed varied between the studies, which also made comparisons between the included studies difficult. For example, some of the studies examining umbilical infections only followed calves for 24 to 72 h. The studies where the calves were followed for limited time ranges made it difficult to draw conclusions because many calves may develop umbilical infections outside of the observed time periods. Furthermore, no studies consistently followed animals from birth to maturity to gain a better understanding of how umbilical infections impact long-term health and productivity.

This review also emphasized the limited amount of research on risk factors for umbilical disease that researchers, and, most importantly, producers, can examine to reduce negative umbilical health outcomes. There is a need to examine more practical risk factors, such as bedding, calving pen cleanliness, calf management, and housing, associated with the development of umbilical infections. This is important because these risk factors can be modified/altered by the producer, whereas risk factors such as genetics are further out of their control. Similarly, there is a lack of clinical research completed on the treatment of umbilical infections. Within this review, only one clinical trial by Abbas et al. [45] evaluated umbilical infection treatment. It is important to expand on this literature to understand ways to effectively treat umbilical disease.

### Limitations of the Review

All search terms were in English, which meant the number of potentially relevant articles excluded for language is not exhaustive. Non-English full texts were excluded due to limited resource allocation to translation services. Furthermore, a large portion of articles were not available, largely due to these articles being older publications. Although many of the excluded articles may not have fulfilled other screening requirements, the combination of any additional articles may have contributed further evidence to this scoping review. Authors should discuss the results and how they can be interpreted from the perspective of previous studies and of the working hypotheses. The findings and their implications should be discussed in the broadest context possible. Future research directions may also be highlighted.

## 5. Conclusions

Through this scoping review, 150 primary research articles were gathered to describe and characterize the literature on umbilical health. An abundance of observational studies examining umbilical health were identified; however, there remains a lack of clinical research, particularly that which evaluates the treatment of umbilical infections. This review further emphasizes the lack of umbilical health definitions and clear methodology reporting, which makes the results of many of these studies difficult to interpret and compare. Therefore, it remains important for researchers to adequately describe all risk factors and outcomes, as well as use reporting guidelines such as REFLECT and STROBE-Vet. These findings suggest that undertaking a systematic review on the one topic of umbilical health in the future may prove challenging, and that additional high-quality umbilical health research is needed to help researchers and shareholders make informed decisions to improve umbilical health in intensively raised cattle.

## Figures and Tables

**Figure 1 vetsci-09-00288-f001:**
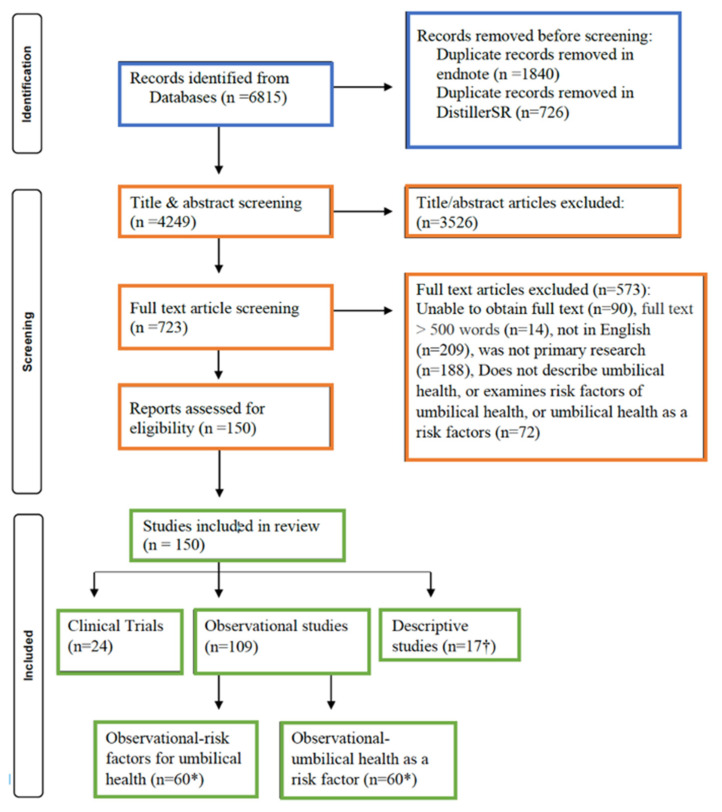
Preferred Reporting Items for Systematic Reviews 827 and Meta-Analysis (PRISMA) flow diagram [35] describing the identification, screening, and inclusion of articles in a scoping review characterizing the literature on umbilical health.* A total of 109 observational articles were included in this review, however, 11 articles contained observational information regarding risk factors for umbilical health and umbilical health as a risk factor and were thus included in both counts and recorded in the tables below, respectively. † Articles that solely contained descriptive statistics were included in this count. Clinical trials or analytical observational studies that contained descriptive information were not included in both counts and represented separately in the tables below.

**Figure 2 vetsci-09-00288-f002:**
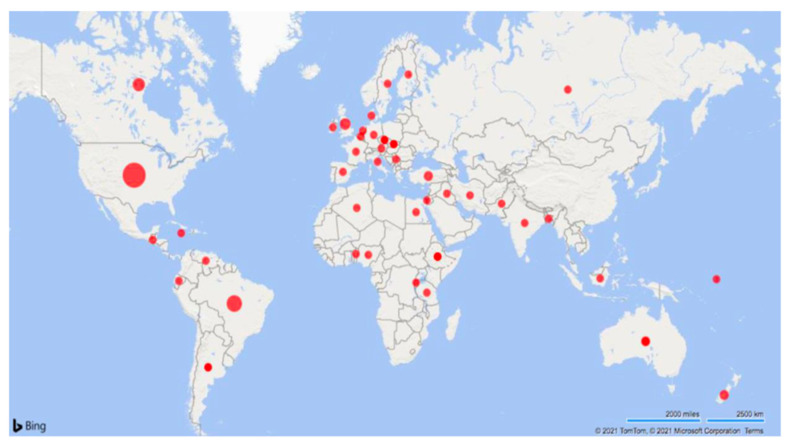
Map of the articles included in the scoping review characterizing umbilical health in intensively raised cattle (1931–2020). The studies (*n* = 109) were completed in the USA (*n* = 41), Brazil (*n* = 23), Canada (*n* = 13), UK (*n* = 10), Australia (*n* = 9), and 36 additional countries that are further described within this review. Larger dots indicate a greater quantity of studies.

**Figure 3 vetsci-09-00288-f003:**
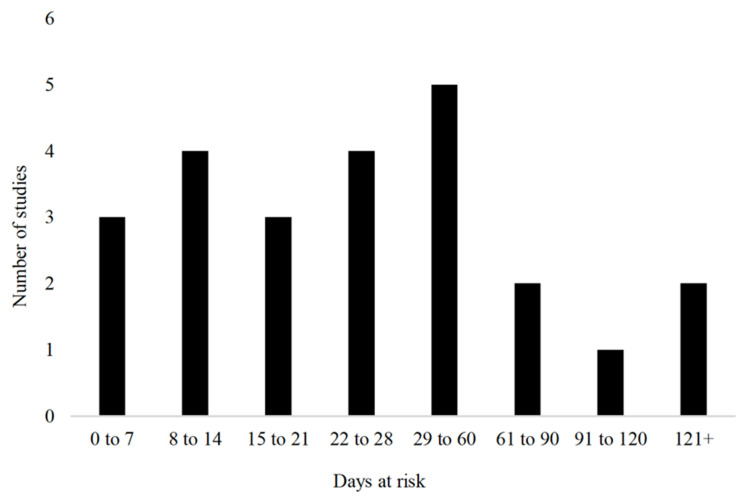
Time at risk (days) for development of outcomes in clinical trials (*n* = 24) included in the scoping review of literature surrounding umbilical health in intensively raised bovines.

**Figure 4 vetsci-09-00288-f004:**
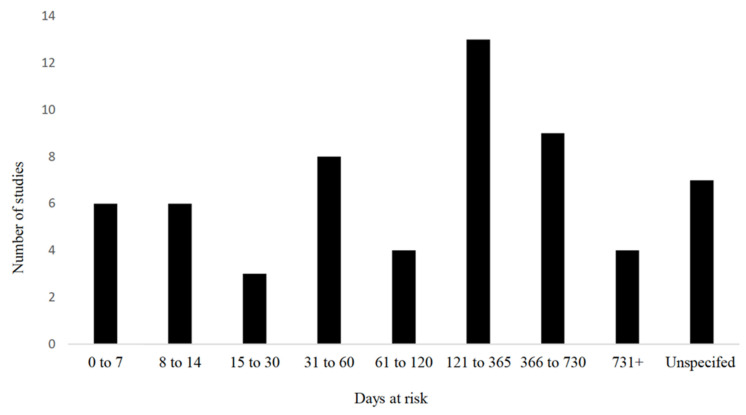
Time at risk (days) for development of outcomes in observational studies examining risk factors for umbilical health (*n* = 60) included in the scoping review of literature surrounding umbilical health in intensively raised bovines.

**Figure 5 vetsci-09-00288-f005:**
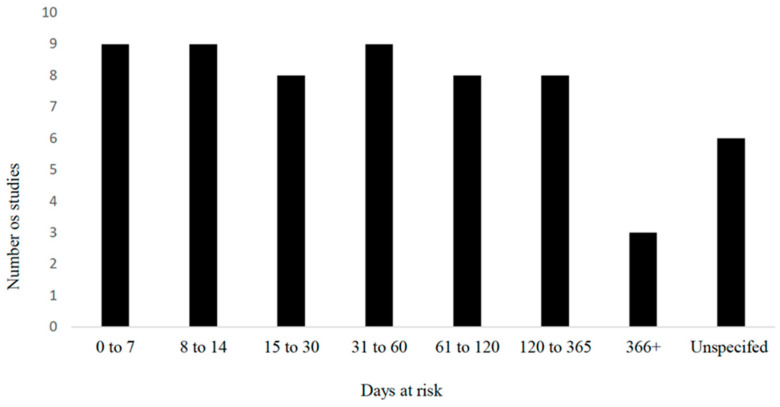
Time at risk (days) for development of outcomes in observational studies examining umbilical health as a risk factor (*n* = 60) included in the scoping review of literature surrounding umbilical health in intensively raised bovines.

**Table 1 vetsci-09-00288-t001:** Initial search results obtained from SCI-EXPANDED, CPCI-S, and ECSI (via Web of Science) conducted on 26 November 2019.

Search	Search Terms	Results
1	(Bovines OR Veal OR Dairy OR Calf OR Calves OR Heifer OR Cow OR Bull OR Sire* OR Steer* OR Beef OR Cow–calf or Dairy-beef OR Feedlot)	759,305
2	(Navel OR Umbilical OR Umbilicus OR Omphal*)	88,675
3	(1 AND 2)	1860

**Table 2 vetsci-09-00288-t002:** Number of articles and date range for descriptive studies, clinical trials, observational studies examining risk factors for umbilical health, and observational studies examining umbilical health as a risk factor.

Study Type	Number of Studies	Date Range
Descriptive	17	1946 to 2018
Clinical Trial	24	1949 to 2018
Observational (risk factors for umbilical health)	60 ^1^	1931 to 2020
Observational (umbilical health as a risk factor)	60 ^1^	1968 to 2020
Total	150	1931 to 2020

^1^ A total of 109 observational articles were included in this review, however, 11 articles contained observational information regarding risk factors for umbilical health and umbilical health as a risk factor and were thus included in both counts and recorded in the tables below, respectively.

## Data Availability

Not applicable.

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
