# Peer review of "Describing and Characterizing the Literature Regarding Umbilical Health in Intensively Raised Cattle: A Scoping Review"

_vetsci, 2022, doi:10.3390/vetsci9060288_

Round 1

Reviewer 1 Report

The manuscript" Describing and characterizing the literature regarding umbilical health in intensively raised cattle: A scoping review" has been written in a good manner with an acceptable arrangement of methodology. I recommend accepting it for publication after the following minor comments: 
1. The title is vague; the manuscript discussed different umbilical affections. I recommend to change "umbilical health" to "umbilical affections" to avoid misunderstanding.
2. Changes in the abstract and other sections should be done according to title change.

Author Response

We thank the reviewer for their time and expertise to review our manuscript.

We are a bit unclear on the meaning of 'affectation' as the reviewer intends - we have found in one medical dictionary that this is defined as disease or morbidity.  We had chosen to discuss 'umbilical health' as opposed to 'umbilical disease' as several outcomes related to umbilical measurements do not necessarily indicate disease - in fact, many outcomes related to umbilical size have not been validated to correspond with disease.  As a result, we would prefer to keep the term 'health' as opposed to morbidity or disease to avoid implying all outcomes captured are as such.

Reviewer 2 Report

Introduction

This focuses on the disease, an aspect, which is covered well and appropriately, but missesthe importance of scientometrics and informatics studies and their potential use in veterinary medicine. Please expand to allow readers to understand the scope and importance of this particular type of work. One paragraph will suffice, as it will enlighten readers about the matter.

Materials and methods

No comments here, the study was designed well.

Only two comments.

1.       During assessment of the articles in how many cases, did you call for a third reviewer? Based on the answer, I might do some further comments on the revised version.

2.       The authors may wish to set a more recent date as the start-point of their work. Really, papers published before 1980 do not offer anything significant in current science. Possibly, they can be included in a separate set, to indicate the historical significance of the work, but that can be all.

Results

Figure 1. Please colourise the boxes by steps to allow readers a better understanding of the steps taken during the study.

Figure 2. Can the authors provide a correlation plot of number of articles published per country versus number of cattle in the respective country?

3.2. – 3.4. Really this part of the manuscript is difficult to read and has important problems. It is just a line of numbers that do not offer much as they are, and they are difficult to read and boring. I fully understand that the results look like this, no problem here. However, I completely disagree with the presentation. Text is not the correct means. I suggest to present all the data in the form of tables, just presenting a brief introductory sentence for each table. This will be  more attractive to future readers.

Discussion

This needs better organisation. For example, there is a section limitations, but still there is a limitation discussed outside that section. Also, different writing styles appear in the text…..

The above show carelessness from the part of the authors.

Please rewrite the discussion carefully and please all authors go through it. I suggest to create three subsections in the discussion.

Finally, please correct Canadian colloquialisms. Please around the world cannot comprehend local parlance of the English language, so I suggest to use British English throughout the manuscript.

Overall. The review is useful and I support publication of this manuscript, but extensive revision as indicated above is needed.

Author Response

We thank the reviewer for their time and expertise to review the manuscript.  We have responded to the comments below (AU:).

Introduction

This focuses on the disease, an aspect, which is covered well and appropriately, but missesthe importance of scientometrics and informatics studies and their potential use in veterinary medicine. Please expand to allow readers to understand the scope and importance of this particular type of work. One paragraph will suffice, as it will enlighten readers about the matter.

AU: We thank the reviewer for this comment, but are unclear as to what study types they wish us to discuss further?  We do not believe there were any informatics studies included, and from our understanding of 'scientometrics' this is another definition for quantitative studies, which would have been everything except for descriptive studies in our results (eg/ clinical trials and analytic observational studies) - can the reviewer please clarify which study types they wish us to comment further on?

Materials and methods

No comments here, the study was designed well.

Only two comments.

  1. During assessment of the articles in how many cases, did you call for a third reviewer? Based on the answer, I might do some further comments on the revised version.

AU:  This was not captured, but is standard practice for synthesis work (see special issue of ZPH focusing on systematic/scoping review methodology (https://onlinelibrary.wiley.com/toc/18632378/2014/61/S1)

  1. The authors may wish to set a more recent date as the start-point of their work. Really, papers published before 1980 do not offer anything significant in current science. Possibly, they can be included in a separate set, to indicate the historical significance of the work, but that can be all.

AU: We appreciate the reviewer's comment and agree that any date set is arbitrary, but this was part of our review question and decided a priori (see our review protocol).  Despite a larger body of work being included than if the date was more recent, the conclusions from the review still indicate that more work in this area is needed, including improvements on standardization of disease and outcome definitions. If the date was more recent, an argument against the same findings could be that more relevant information was missed.  We feel the 1980 date shows that despite a large body of work included, this are of research requires more attention to determine efficacy of prevention and treatment strategies, as well as improved research methodology with standardized outcome definitions.

Results

Figure 1. Please colourise the boxes by steps to allow readers a better understanding of the steps taken during the study.

AU: Amended

Figure 2. Can the authors provide a correlation plot of number of articles published per country versus number of cattle in the respective country?

AU: We appreciate the reviewer's comment but don't feel this will add value to the manuscript, from the existing figure we feel we have highlighted where the studies in our review came from, and the reader can appreciate that this is not comprehensive globally, and that many important regions have few articles. We will defer to the editor.

3.2. – 3.4. Really this part of the manuscript is difficult to read and has important problems. It is just a line of numbers that do not offer much as they are, and they are difficult to read and boring. I fully understand that the results look like this, no problem here. However, I completely disagree with the presentation. Text is not the correct means. I suggest to present all the data in the form of tables, just presenting a brief introductory sentence for each table. This will be more attractive to future readers.

AU: We appreciate the reviewer's comments here, but with this not being noted in the other reviewers comments we would like to defer to the editor.  We have included several figures and graphs to break up the data, but feel that presenting all data as figures and graphs would become unwieldy for the reader and make the already lengthy manuscript substantially longer.

Discussion

This needs better organisation. For example, there is a section limitations, but still there is a limitation discussed outside that section. Also, different writing styles appear in the text…..

AU: We appreciate the reviewers comment but would appreciate line numbers to identify what areas they feel are in the incorrect section.  There is a 'limitation' discussed in line 592 onward, but this is specific to a limitation of the body of literature - the limitation section (4.1.) refers to limitations of the review - limitation of the body of literature is essentially results, which is why it is discussed earlier.  If this is the issue, we have amended section 4.1. to be titled "Limitations of the review" to be clear this section pertains to the review methodology, and that limitations of the body of literature are the results of the review and discussed in the main body of the discussion.

The above show carelessness from the part of the authors.

AU: We thank the reviewer for the comment, but after reading the discussion would appreciate line comments to identify where they feel there are problems with the writing style.

Please rewrite the discussion carefully and please all authors go through it. I suggest to create three subsections in the discussion.

AU: Thank you for this comment - we are unclear what 3 subsections the reviewer would suggest?  Our discussion centres around five main points: (1) lack of standardize outcome definitions (and suggestions towards standardization),(2) lack of clear reporting/use of reporting guidelines in primary research, (3) lack of intra- or inter- observer reliability testing for subjective outcome measurements, (4) a wide range of disease incidence (and discussion on several factors that may contribute), and (5) research gaps.

Finally, please correct Canadian colloquialisms. Please around the world cannot comprehend local parlance of the English language, so I suggest to use British English throughout the manuscript.

AU: We appreciate this feedback but would ask the reviewer to provide line numbers for the areas that contain colloqualisms, as we do not know what aspects of the language of the manuscript they are referring to.

Overall. The review is useful and I support publication of this manuscript, but extensive revision as indicated above is needed.

Reviewer 3 Report

The manuscript offer a thorough description of the umbilical health conditions found in the available literature. 

The meta analysis is well structured and a good description of the findings are included.

Some minor comments have been included within the PDF attached.

Author Response

We thank the reviewer very much for their time and expertise to review the manuscript, and have made changes to the submission (as highlighted) and have responded to comments within the attached PDF.

Round 2

Reviewer 2 Report

Minor revision.
The authors have really evaded to answer the comments made during the initial review. I fully understand the situation and the circumstances, hence no problem at all.
Continuing within the same situation and circumstances, now I suggest to correct the various language mistakes scattered across the text.